# Prebiotic Organic Chemistry of Formamide and the Origin of Life in Planetary Conditions: What We Know and What Is the Future

**DOI:** 10.3390/ijms22020917

**Published:** 2021-01-18

**Authors:** Bruno Mattia Bizzarri, Raffaele Saladino, Ines Delfino, Juan Manuel García-Ruiz, Ernesto Di Mauro

**Affiliations:** 1Ecological and Biological Sciences Department (DEB), University of Tuscia, Via S. Camillo de Lellis snc, 01100 Viterbo, Italy; bm.bizzarri@unitus.it (B.M.B.); delfino@unitus.it (I.D.); dimauroernesto8@gmail.com (E.D.M.); 2Laboratorio de Estudios Cristalográficos, Instituto Andaluz de Ciencias de la Tierra, Consejo Superior de Investigaciones Científicas–Universidad de Granada, Avenida de las Palmeras 4, Armilla, 18100 Granada, Spain

**Keywords:** formamide, prebiotic chemistry, geochemistry, biomorphs, origin of life

## Abstract

The goal of prebiotic chemistry is the depiction of molecular evolution events preceding the emergence of life on Earth or elsewhere in the cosmos. Plausible experimental models require geochemical scenarios and robust chemistry. Today we know that the chemical and physical conditions for life to flourish on Earth were at work much earlier than thought, i.e., earlier than 4.4 billion years ago. In recent years, a geochemical model for the first five hundred million years of the history of our planet has been devised that would work as a cradle for life. Serpentinization processes in the Hadean eon affording self-assembled structures and vesicles provides the link between the catalytic properties of the inorganic environment and the impressive chemical potential of formamide to produce complete panels of organic molecules relevant in pre-genetic and pre-metabolic processes. Based on an interdisciplinary approach, we propose basic transformations connecting geochemistry to the chemistry of formamide, and we hint at the possible extension of this perspective to other worlds.

## 1. Introduction

Any theory for the origin of life “as we know it” must be geochemically plausible. So far, this geochemical constriction has not been carefully considered because we have a poor and rather vague knowledge of the lifeless Earth, i.e., of the geological framework in which life arose. In fact, unveiling what our earliest, lifeless planet was like is a formidable challenge. The very name that geologists have given to that time, the Hadean (a Greek term that refers to hell) is because the surface of the planet at that time was thought to be a hell, i.e., a dry, inhospitable world of extreme conditions, of intense ultraviolet radiation, high temperature, no liquid water, covered by incandescent volcanoes and magma seas, and frequently bombarded by bolides. Therefore, it was thought that for the first million years, the surface of the Earth was not only an uninhabitable scenario where “life as we know it” could not either appear or survive, but even the prebiotic chemistry reactions needing a solvent such as water would not develop [1,2,3]. However, this drastic view has radically changed in recent years, thanks to the information inferred from preserved zircon crystals that formed during the Hadean, demonstrating the presence of liquid water 4.4 billion years ago [4]. This discovery has changed our geochemical view of the infancy and the youth of the planet. It has also allowed a new timing for the beginning of prebiotic chemistry and the origin of the first molecular architectures that could be considered living organisms. The geochemical testing/validation of any proposed chemical pathway intended to explain how life appeared on this planet is now unavoidable. Starting from the new information provided by the crystals of zircon, we have devised a geochemical model for the first six hundred million years of the history of our planet [5]. Water condensed on the surface of the planet before 4.4 billion years ago, i.e., soon after solidification of the first outer surface of an undifferentiated mantle, the first ultramafic rock crust. Therefore, it is very likely that the water would interact with iron magnesium silicates, triggering serpentinization. This unavoidably created an alkaline hydrosphere and a reduced atmosphere rich in H_2_ and CH_4_. The crust of the planet under these physicochemical conditions should have been a global factory of simple and complex organic compounds. Furthermore, once the first felsic protocontinents emerged among the alkaline seas, the high pH waters became enriched in silica and provided the chemistry to trigger the formation of mineral self-organized structures (MISOS) [5,6].

### 1.1. A Hospitable Hadean Time Factory of Organic Synthesis

The rationale behind the model described above of a hospitable Hadean time has been exposed [5]. We just need to outline two consequences of serpentinization that are relevant for the role of prebiotic chemistry of formamide in the Hadean time. The first is the release of H_2_, which in contact with CO_2_ expelled by the inner planet created a reducing atmosphere rich in methane [7]. Because of the higher geothermal gradient at that time, the serpentinization reaction should have been easily associated with the Fischer–Tropsch reaction, known to yield polycyclic aromatic hydrocarbons and other organic molecules, catalyzed by magnetite forming during the serpentinization reaction. While other abiotic sources of organic compounds exist [8,9], serpentinization appears to be the most efficient geochemical source of organic synthesis. Scheme 1 reports representative examples of serpentinization (Panel A) and Fischer–Tropsch (Panel B) reactions.

Beyond the direct contribution of organic molecules, the main impact of the serpentinization reaction is the creation of a Hadean reducing atmosphere that, perhaps catalyzed by lightning and ultraviolet and photonic radiation, would become a laboratory for the synthesis of organic molecules of higher complexity and prebiotic interest. It is not known for how long the reducing methane-rich atmosphere generated by large-scale serpentinization lasted. But the transition rate to a CO_2_ rich atmosphere was probably slow, allowing this natural laboratory to work for millions of years. It has been demonstrated that Miller-type experiments in a reduced atmosphere such as the one claimed here for the Hadean synthesized most of the bricks of life, including hydrogen cyanide (HCN) and formamide (HCONH_2_). It has been argued that liquid HCONH_2_ could have been accumulating in the early ocean through the dissociation of ammonium formate at about 180 °C in the presence of mineral catalysts, a compound that could have formed by the gas-phase reaction of formic acid with ammonia [10]. Interestingly, these experimental studies also suggest that minerals such as Fe–Ni thiospinel catalyze the Fischer–Tropsch type (FTT) conversion of carbonate to formate, an alternative but parallel route to HCONH_2_ production in a Hadean geochemical scenario proposed in our model, and that events associated with terrestrial volcanism could result in the accumulation of formic acid, the acid constituent of HCONH_2_. All in all, it seems that in the factory of synthetic organic molecules created in the early Hadean Earth, formamide was one of the main products. The factory was equipped with a reducing atmosphere, with lightning, volcanism, geysers, and the earliest hydrothermal systems pumping water laden with FTT reaction products to the surface. It is unknown for how long this reduced atmosphere with high CH_4_/CO_2_ ratio lasted. It was certainly a transition atmosphere progressively enriched in CO and CO_2_, as the production of H_2_ decreased with the exhaustion of fresh ultramafic rock for serpentinization.

### 1.2. The Role of Minerals and Mineral Self-Organization in Prebiotic Chemistry

The second important output of the serpentinization reaction is the production of a high concentration of hydroxyl so that the waters of the primitive lakes and oceans were alkaline and compatible with HCONH_2_ condensation and with soda lakes of the African Rift Valley [11,12]. When the first felsic rocks, granites and granitoids, started to emerge from the alkaline oceans they became enriched in silica (pH 12.2). There are no data on the pH value resulting from serpentinization reactions occurring under early Hadean reducing atmospheric conditions, i.e., at low concentrations of CO_2_, but they probably reached a higher pH value. At these high pH values, the concentration of silica in the Hadean seas and lakes could reach up to several grams per liter, mobilizing silica at a large scale and explaining the ubiquitous presence of silica and silicified rocks [5]. Both thermodynamics and kinetical reasons sustain the idea that minerals play a crucial role in the complex chemical routes to the first living organisms. The soup of Darwin’s little pond [13,14] works much better with some pasta offering surface to reactions. From the initial bet of Bernal for montmorillonite, other minerals have been proposed to speed up the formation of molecules required for life and were suggested to play as epitaxy substrates and nanoreactors [15,16].

Probably inspired by the studies of modern industrial catalysis, most of the studies so far reported on the role of metals in origin of life focused on transition metals and their oxides and hydroxides (see Table 1). However, in spite of the ability of silica to adsorb and interact readily with organic matter, the role of silicon and silica has been much less explored. Interestingly, the alkaline silica-rich lakes and oceans bring to the scene fascinating mineral self-organized structures (MISOS) induced by silica, namely (a) silica/carbonate nanocomposites with biomimetic morphology and textures [17]; and (b) mineral membranous vesicles made of silica and metal oxy-hydroxides. Both of them form under extreme conditions of high pH and high silica concentration. These are nowadays bizarre conditions only fitted by spring waters linked to contemporary serpentinization. However, during the Hadean, rather than extreme, these conditions would have been widespread, regular environmental conditions. MISOS have been demonstrated to be geochemically plausible [18]. Osmotic driven mineral membranes are particularly important in the context of prebiotic chemistry (see below). At later stages, when the level of atmospheric CO_2_ increased, it dissolved in these alkaline waters, and the Hadean seas and lakes became carbonate-rich, similar to modern soda lakes [19]. These experiments further support the compatibility of the formation of HCONH_2_ and its efficient condensation with the geochemical environment depicted in our view of the Hadean.

## 2. Prebiotic Organic Chemistry and Chemical Complexity

We consider that self-organization and systems chemistry processes of pristine biogenic molecules played a relevant role in the emergence of life. Multi-component reactions (MCRs) provide the tool for the formation of large libraries of organic compounds by one-pot sequential combination of simple reagents. MCRs involve a minimum of three reactants or reaction centers and are conducted in one pot and with a single operational step [20]. HCN and HCONH_2_ are both effective substrates in MCRs. They are easily transformed into reactive intermediates that react combinatorially [21]. HCONH_2_ is liquid between 6 °C and 210 °C and has a high dielectric constant, which makes it a solvent for polar biomolecules, favoring successive oligomerization [22]. HCONH_2_ is produced by several reaction pathways in Earth-like conditions, including electric discharge [23], radiolysis processes [24], and plasma impact events [25]. The local concentration of HCONH_2_ is increased by geochemical processes, such as water evaporation in drying lagoon, adsorption in clays, silica, TiO_2_, thermophoresis in hydrothermal micropores, and dry/wet heating/cooling cycles [26,27]. Interaction of HCONH_2_ with minerals affects the fate of biomolecules, i.e., the stability of oligonucleotides, enhancing the chemical complexity and informative content of the system [28,29]. Minerals are catalysts for HCONH_2_, determining pathways, energetics, and kinetics of the prebiotic processes [30,31]. Table 1 reports the class of biogenic molecules synthesized from HCONH_2_ and minerals in different energy conditions, encompassing volcanic-like environments, high energy radiative conditions, electric discharge, shock wave and plasma impacts. The inventory of chemical complexity is impressive, including intermediates of the pre-genetic and pre-metabolic apparatuses [32]. The different types of molecules are produced contemporaneously, limiting the spatial distribution of the products [33]. Terrestrial and non-terrestrial minerals act as catalysts [34,35]; the reactions also working in the presence of H_2_O [36]. Unified mechanisms involving neutral and radical species can account for the formation of the reported biomolecules, and HCN oligomers have been identified as key intermediates in the reaction pathway [37]. The mineral surface is also responsible for the control of the selectivity of the reaction, as in the case of the synthesis of β-nucleosides from sugar and nucleobases [38].

The role of HCN in the synthesis of nucleobases was explored in thermal conditions. Canonical nucleobases adenine and guanine, and other biogenic purine derivatives, such as xanthine, hypoxanthine and 2,6-diaminopurine, were obtained from HCN [61,62] and NH_4_CN [63]. In long-term experiments (0.5 to 37 years) in solution and at low temperature (−78 °C), uracil was also detected [64]. The possibility of HCN conversion to other reagents, such as HCONH_2_ and HCOONH_4_, was not considered.

### 2.1. Alternative C-1 Chemical Precursors: The Gas-Chemistry

Methane and carbon oxides are alternative reagents for the synthesis of biogenic molecules in electric discharge experiments. The pioneering Miller–Urey experiment [65] provided data about the formation of amino acids and carboxylic acids by MCC. Table 2 reports the products by electric discharge in different models of pristine atmosphere. These models evolved from a reducing to a more neutral atmosphere, according to geochemical data.

Amino acids and carboxylic acid derivatives were produced by a Strecker-like condensation involving HCN and carbonyl derivatives. Computational simulation of the Miller–Urey experiment suggested the formation of HCONH_2_ as a key intermediate in the synthesis of glycine, and HCONH_2_ was isolated in a successive experiment [72]. In this latter case, in addition to the expected amino acids, the four nucleobases were synthesized.

### 2.2. The Need for Catalyst Complexity

How can the level of the catalyst complexity affect the reaction pathways? A clear-cut case is that of meteorites. Table 3 reports the product-to-product relationships for the reaction of HCONH_2_ and 19 meteorites of different types by the pair-wise scatter plots and Pearson correlation analysis [73]. The data are the frequency by which a class of biogenic molecules is synthesized with respect to a reference counterpart. The Pearson correlation coefficient was calculated for each considered couple of products and the corresponding correlation probability (P) was calculated. The reactions computed are referred to three energy sources: (i) thermal condensation (condition B); (ii) thermal condensation in HCONH_2_/H_2_O mixture (condition C); and (iii) high energy proton beam irradiation (condition D). The data used for the analysis (that is the yield of reaction products in relation to the specific experimental conditions) were extracted from references [51,52,53,57], describing the prebiotic chemistry of HCONH_2_ in the presence of meteorites. A high value of the correlation probability between the relative yields of reaction products was observed for the synthesis of purine and pyrimidine nucleobases, further extended to the formation of amino acids, low molecular weight compounds (LMWC) and condensing agents. High molecular weight compounds (HMWC) were not correlated, or alternatively, they were negatively correlated to the other families of products. Considering the energy source, purine and pyrimidine nucleobases, and amino acids and condensing agents, are significantly correlated under condition D (Table 3). Purine nucleobases showed a high value of correlation with amino acids in condition B, and with condensing agents in condition C. In this latter case, a high level of correlation was observed between amino acids and carboxylic acids (both LMWC and HMWC). Amino acids are positively correlated to LMWC, while they are negatively correlated to HMWC. These observations pointed to an interplay between amino acids and carboxylic acids in condition C, which depends on the complexity of the carboxylic acid. The production of HMWC is somehow disfavored by that of amino acids and vice versa. In this frame, it is interesting to note that there was a negative correlation between LMWC and HMWC in C conditions. Thus, the production of low molecular weight acids seems to be in competition with the production of HMWC.

Notably, the indication of the interplay between HMWC and LMWC was opposite (with a low correlation) when the production occurred under irradiation (condition D). In the C condition, a medium negative correlation occurred for HMWC and condensing agents. These data suggest that in the presence of meteorites, the selectivity of the reaction is not dependent on the composition of the catalyst per se, since general patterns are operative from HCONH_2_ independently from the nature of the meteorite. All types of meteorites catalyze the formation of biomolecules, the thermal scenario being slightly more effective than the radiative counter-part in the formation of pre-metabolic ingredients, nucleobases prevailing in the presence of the proton beam. The negative correlation of HMWC with the other biomolecules suggests the presence of unfavorable conditions for the synthesis of the building blocks of organic-based membranes from HCONH_2_ and meteorites, highlighting the possibility that inorganic compartmentalization plays a role in molecular evolution.

### 2.3. The Emergence of a Catalyst-Driven Compartmentalization in HCONH_2_-Based Geochemical Scenarios

Compartmentalization is a fundamental process for life, allowing the dynamic separation of the internal volume of the cell and the external medium. Chemical systems that spontaneously generate compartmentalization are models of pre-cellular systems [74]. From the bottom-to-up point of view, compartmentalization occurs as a consequence of two steps: (i) the synthesis of appropriate building blocks; and (ii) the self-assembly of building blocks into a membrane-like structures. The components of the pre-cell form independently inside the membrane, which may promote a network of chemical transformations. Mineral self-organized structures (MISOS) induced by silica can reverse this perspective, especially so for silica-metal oxyhydroxide membranes. These membranes are composed of two layers, one of amorphous silica and another of metal oxide-hydroxide nanoparticles and has been shown to form under the geochemical conditions of the Hadean. Their formation entails the creation of a compartmentalization of space via vesicles and tubular structures [16]. This ionic exchange generates electrostatic potential up to the scale of hundreds of millivolts that remains active for several hours [75], and is able to convert the structure into activated reactors. MISOS showed a clear catalytic effect when growing in the presence of HCONH_2_, yielding a large panel of biogenic molecules, nucleobases, amino acids and carboxylic acids prevailing inside of the membrane with respect to the outside counterpart [56]. Therefore, our alkaline model is the first evidence for a global scale geochemical scenario in which the silica produced from serpentinization provides the reaction medium for the catalytic compartmentalization process, affording both inorganic membranes and biogenic compounds. The experiments performed so far reveal a relevant role for metal membranes. Detailed investigation of the membrane nanostructures to account for their catalytic properties has only been performed for iron under oxidation conditions. These experiments must be performed under the reduced conditions of the Hadean. Certainly, the possibilities of the geochemical scenario are not yet fully exploited by our experiments, and a higher level of complexity can be reached. MISOS deserve further consideration. MISOS are produced at the microscale level by addition of nano- to micro-liter drops of common metal salts to alkaline silica-rich solutions similar to those deriving from serpentinization, mimicking geochemical environments more realistic than chemical gardens made with pellets of salts [54]. When MISOS are produced in the presence of HCONH_2_, the catalysis-driven compartmentalization process starts immediately, and a large panel of biogenic molecules is produced. The vesicles show an average diameter of 250 microns and grow to reach a critical size, beyond which they break down releasing their content in the bulk of the solution [56]. In a general scenario in which vesicles form and break in sequence, the content of molecules of the first series can be incorporated into the second and so on, resulting in a continuous increase in molecular content. As for biomorphs of silica-carbonate, their catalytic properties have not been yet fully investigated. Preliminary experiments indicate selective absorption of amino acids, allowing the formation of hybrid composites that could extend the complexity of mineral self-organization. We believe that it is time for well-designed rock–fluid interaction experiments.

## 3. Prebiotic Chemistry Meets Geochemistry: How Can the Global Scale/Formamide Model Be Extended and Validated?

The terrestrial model may be extended towards the search for the conditions causing and allowing the development of higher-complexity chemical species. The most advanced system is that of nucleic acid components, in which the formation of nucleosides, their phosphorylation, their cyclization, and their polymerization have been reported [76]. Could the HCONH_2_ chemistry reach this complexity? The other very interesting possibilities concern amino acids and carboxylic acids, whose components are present in the biomorph chemistry as products, but only at an initial level of complexity. The open question at this stage is: can this complexity be increased?

Figure 1 reports the tricarboxylic acid (rTCA) cycle (Krebs cycle) considered the prebiotic “core of the core” of metabolism [77,78,79]. The complexity of the Krebs cycle resides in the sequential transformation of different carboxylic acids with concomitant formation of other biomolecules. Biomorphs catalyze the synthesis of some intermediates of the Krebs cycle from HCONH_2_ (Figure 1, color code red). Other minerals, such as TiO_2_, and meteorites afforded a larger panel of intermediates, almost closing the whole cycle, while others were less effective. Thus, conditions may exist in which the ensembles of different minerals may interact synergically to increase the chemical complexity. In principle, no specific limits are expected if adequate reagent, catalysts and energy are available. For this to occur, an evolutionary advantage must be involved. In abiotic conditions, possible advantages could consist in: acquired stability of the products (i.e., higher stability of polymers relative to monomers), an increased fitness to the environment (i.e., higher solubility or solubility in different solvents), onset of cyclicity of the reactions, and reciprocal stabilization. These models are in principle universal. This observation opens the way to the consideration that further progress toward complexity is channeled by the specific environment in which these reactions occurred. If on Earth the “biomorph and silica” scenario prevailed because of the global scale and generality of these reactions, allowing the contemporaneous synthesis of the biogenic panels described above, in celestial bodies with a different geological history, the panels produced could have been different, leading to different local pre-LUCA systems [80,81]. Regardless, the facts that HCONH_2_ condensation always affords biomolecules, and that “exotic” compounds are essentially rare and limited, claims that the forms of chemical complexity that will eventually become biology are not likely to be basically different from the only one we know.

## 4. Other Worlds

Is the geochemical scenario triggered by the serpentinization reaction unique to the Earth, or can it be extrapolated to other celestial bodies? Was this factory of organic molecules privative to the Earth, or did it work, is working, or will work in different places of the universe? The answer is easy to prospect. In principle, serpentinization should have been at work in any other Earth-like planets and moons, but also in meteorites, asteroids, comets, and even in the cosmic dust. It is evident that a *conditio sine qua non* for serpentinization to occur is the presence of water. Nevertheless, the markers that would allow one to identify these factories of organic molecules in extraterrestrial bodies are mainly three, namely: chemical precursor, alkaline seas/lakes, and carbonate evaporites. The first would indicate ongoing serpentinization processes. The second one would be characteristic of the late stages of the atmosphere’s evolution. There are other markers, but they are more challenging to detect with remote sensing technologies. Considering our two closest neighbor planets, we can rule out Venus, where neither the presence of methane nor any type of signal typical of an alkaline or carbonate environment has been detected. Its acidic atmosphere instead suggests the opposite. However, the geological infancy of Mars and Earth are considered to be so similar that it would be very strange that serpentinization processes did not dominate the Noachian geochemistry of the red planet. Several findings support this idea, including the detection of hydrated minerals, such as serpentine phases, water-laid sediments and hydrothermal sinter deposits [82,83]. Therefore, it is entirely possible that important steps in prebiotic chemistry, in mineral self-organization processes, and in early evolution of life were preserved in these Martian deposits. When searching for serpentinization-driven geological scenarios elsewhere, it will be a mistake to seek geological landscapes similar to the one we have depicted for our Hadean Earth [84]. How the serpentinization reaction develops depends very much on the planet’s size or moon and the distance to their stars, and other variables. Many of these geological structures and processes might have never occurred on Earth. Thus, highly alkaline environments likely related to serpentinization have been suggested to exist on different moons of the solar system, including Enceladus, Europa, Ganymede, and Titan. The evidence is in some cases supported by the large amounts of organic compounds, as in the case of Titan [85]. In the case of Enceladus, silicate minerals have been detected in the dust released by their plumes [86].

The extremely high concentration of organic compounds in the atmosphere of the moon of Saturn could likely be produced by an ongoing planetary-scale factory based on a subsurface ocean serpentinization. The asteroid’s belt could also be a place to look for. Ceres, the larger object in the belt has two of the features of a late stage of serpentinization-driven process: a crust made of ice-water with about 20% of organic compounds and spots of large amounts of sodium carbonate on the surface [85,86,87,88]. In general, it can be reasonably argued that any celestial body containing magnesium-iron silicates is a niche for serpentinization, mineral self-organization and organic synthesis. How large and diverse the panoply of organic molecules is depends on the mineral composition, the source of energy available and the external physical parameters, a subject to be developed elsewhere.

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
