# Peer review of "Prebiotic Organic Chemistry of Formamide and the Origin of Life in Planetary Conditions: What We Know and What Is the Future"

_ijms, 2021, doi:10.3390/ijms22020917_

Round 1

Reviewer 1 Report

The article of R. Saladino and collaborators is the nth article where formamide is descibed as a valuable source of carbon for the developing of life as we know on our planet and maybe in or solar system. The work, in the form of a review is a limited contribution in the field of prebiotic chemistry and geochemistry/geology marriage.

I sent in attachment for the authors a pdf file where I noted all the correction needed, concerning the preparation of the manuscript, adding some equation and additional figures. Here and there some typos errors, and some comments on the too much self-citation typical of some Italian researchers. I am not a native English speaker or mother tongue and I relatively feel qualified for the corrections. This is a review and it need to be innovative in the field, while thousands of report were already published in the field.

Reviewer 2 Report

Summary

The Review paper from Saladino and coworkers discusses the potential role of formamide during the early period of chemical evolution taking place on the Earth, including its production, transformations and role as a solvent.  Saladino is well-known for his investigations of formamide, and this manuscript provides an up-to-date account of a likely geochemical scenario that could have been sustained over geological time scales that would have generated formamide, namely through serpentinization.  It is well-known that Miller-type syntheses of organic compounds are most efficient under reducing conditions, however, it is likely that core/mantle differentiation of the Earth resulted in at atmosphere dominated by CO2 and N2 early in its history. The authors argue for the scenario that global-scale serpentinization would have taken place concurrently with condensation of the hydrosphere by the interaction of liquid water with ultramafic rocks providing a reducing atmosphere possibly for millions of years.  One of the main products of a reducing atmosphere would have been formamide, which can be used as a reagent/solvent for making a variety of prebiotically relevant compounds.  In addition, serpentinization results in alkaline waters, which could have resulted in mineral self-organized structures based on silica and/or carbonate.  Such mineral structures could have served as important surfaces or compartments for prebiotic chemistry. The reviewer recommends publication after the following criticisms have been resolved.

Line 72: The authors mention that ammonium formate can serve as a precursor for formamide, citing reference 6 which cites another reference (https://www.sciencedirect.com/science/article/pii/S092633730800372X?via%3Dihub#aep-section-id27). This reaction of ammonium formate to produce formamide involves heating it presumably in the absence of water within a relatively narrow range of temperatures.  It is questionable how relevant this reaction is under relevant prebiotic conditions.  Could the authors please cite additional prebiotically relevant references to back up this claim, or at least note the exact conditions in which these experiments were carried out?

The global presence of ultramafic rocks during the condensation of the hydrosphere after the moon-forming impact is crucial for the proposed scenario.  Since I presume most people reading this review will not necessarily be experts in geology (including myself), could the authors please provide an explanation and a reference as to why ultramafic rocks as opposed to felsic rocks are expected at the surface? Mafic rocks are more dense, so would they not be expected to be below the felsic rocks? Why did more feslic rocks then appear later?

Table 1, Entry 9. Nucleosides are claimed amongst the products, but that is misleading because that cited paper is about phosphorylation of nucleosides to nucleotides using formamide as a solvent.  In fact, Table 1 would be greatly improved by adding an extra column that clearly states formamide’s role as a solvent or reagent. 

Line 118: The term multi-component chemistry is a good one, and is one the reviewer has not explicitly heard before in the context of prebiotic chemistry.  Could the authors please provide a brief definition and citation if possible? Is this term meant to be at all different from the more often used term in organic chemistry, i.e., “multi-component reaction”?  Is it worth while to distinguish between "multi-component chemistry" and "multi-component reactions"? 

Line 134: While many products can be formed at once, it is not known if such complex mixtures are actually what is needed for life to arise.  Spatially separated reactions, which later combine, e.g., for furnishing nucleosides, are beneficial for obtaining what are perceived to be reasonable yields.  The authors should note that whether some degree of spatial separation is required for some key steps/processes in prebiotic chemistry is unknown.  In the opinion of the reviewer, understanding how complex chemistry leads to life-like processes is the next frontier in OOL research, as there are now many known abiotic routes to supposedly relevant biomolecules.   

Line 143: HCN chemistry has resulted in more purines than just adenine and guanine.  Please see the following reference. https://www.mdpi.com/2075-1729/3/3/421

Section 2.2.  This is an interesting section, but only one reference is cited (63), but has nothing to do with the actual chemistry discussed.  How exactly did the authors carry out this study?  Is this analysis unique to this review, or is there a reference cited which can point the reader where to read more?  If unique to this review, the authors need a more detailed explanation for how this analysis was actually carried out and what specific references were the source of data for the analysis.

Section 3.  This is also an interesting discussion, but likewise, the relevant references are not afforded to the reader.  In particular, at least Figure 1 should contain all the references needed to justify/provide evidence for each colored line.

Line 286 needs a reference.

Line 297 needs a reference.

Line 302 needs a reference.

Minor edits:

Line 33: delete “the” before “hell”

Line 34: insert “i.e.” between “hell,” and “a dry”

Line 37: change “inhabitable” to “uninhabitable”

Line 38: delete the hyphen before “needing”

Line 41: change “million-year” to “billion years”

Line 45: change “in” to “on”

Line 47: the Earth is not that old, please change to a more accurate age.

Line 58: replace “reach” with rich

Line 58: delete “the”

Line 94: insert “to” between “up” and “several”

Line 96: add “the” after “to”

Line 109: add “been” after “have”

Line 233: change “full” to “fully”

Line 250: delete “the” before “metabolism”
